# Epigenetic Modifiers to Treat Retinal Degenerative Diseases

**DOI:** 10.3390/cells14130961

**Published:** 2025-06-23

**Authors:** Evgenya Y. Popova, Lisa Schneper, Aswathy Sebastian, Istvan Albert, Joyce Tombran-Tink, Colin J. Barnstable

**Affiliations:** 1Department of Neuroscience, Penn State University College of Medicine, Hershey, PA 17033, USA; eyp1@psu.edu; 2Department of Molecular and Precision Medicine, Institute for Personalized Medicine, Pennsylvania State University College of Medicine, Hershey, PA 17033, USA; lschneper@pennstatehealth.psu.edu; 3Department of Biochemistry and Molecular Biology, Huck Institute of Life Sciences, Eberly College of Science, The Pennsylvania State University, University Park, PA 16802, USA; azs13@psu.edu (A.S.); iua1@psu.edu (I.A.); 4Skyran Biologics, Harrisburg, PA 17112, USA; jttink@aol.com

**Keywords:** retina degeneration, chromatin, epigenetic, rod photoreceptor

## Abstract

We have previously demonstrated the ability of inhibitors of LSD1 and HDAC1 to block rod degeneration, preserve vision, maintain transcription of rod photoreceptor genes, and downregulate transcripts involved in cell death, gliosis, and inflammation in the mouse model of Retinitis Pigmentosa (RP), rd10. To extend our findings, we tested the hypothesis that this effect was due to altered chromatin structure by using a range of inhibitors of chromatin condensation to prevent photoreceptor degeneration in the rd10 mouse model. We used inhibitors for both G9A/GLP, which catalyzes methylation of H3K9, and EZH2, which catalyzes trimethylation of H3K27, and compared them to the actions of inhibitors of LSD1 and HDAC. All the inhibitors are likely to decondense chromatin and all preserve, to different extents, retinas from degeneration in rd10 mice, but they act through different metabolic pathways. One group of inhibitors, modifiers for LSD1 and EZH2, demonstrate a high level of maintenance of rod-specific transcripts, activation of Ca^2+^ and Wnt signaling pathways with the inhibition of antigen processing and presentation, immune response, and microglia phagocytosis. Another group of inhibitors, modifiers for HDAC and G9A/GLP, work through upregulation of NGF-stimulated transcription, while downregulating genes belong to immune response, extracellular matrix, cholesterol signaling, and programmed cell death. Our results provide robust support for our hypothesis that inhibition of chromatin condensation can be sufficient to prevent rod death in rd10 mice.

## 1. Introduction

The epigenetic landscape of a cell plays a major role in controlling patterns of gene expression rather than being a passive response to other mechanisms. All cell nuclei contain both euchromatin, which is relatively open and can be actively transcribed, and heterochromatin, where the nucleosomes are more densely packed and transcription is minimal. The transition between these chromatin states is dynamic and under multiple forms of control. Cells with open euchromatin organization can survive changes in cellular homeostasis in response to stress, but these types of cells are also predisposed to malignancy. Cells with more heterochromatic organization are less susceptible to cancerous transformation but have less ability to adapt to changing environments, which makes them prone to cell death or degeneration. Mature mouse rod photoreceptors, like most neuronal cells, belong to the second group of cells and additionally have a uniquely closed chromatin organizational structure [1,2,3]. In murine rod photoreceptors, chromatin forms layers with distinctive patterns of posttranslational histone modifications. Constitutive heterochromatin is found in the center of the nuclei marked by H3K9me3 and H4K20me3. During rod maturation, a large-scale gene repression occurs, creating a layer of facultative heterochromatin around the central aria marked by H3K27me3 and H3K9me2. The euchromatin layer, containing active genes marked by histone acetylation and H3K4me2/3, is located at the nuclear periphery [1,2,4]. The restricted transcription potential of metabolically active rod photoreceptors may increase efficiency under normal conditions but may make them less able to respond under conditions of stress.

Among the retinal degenerations that cause severe vision loss, Retinitis Pigmentosa (RP) affects 1.5 million people worldwide. It is inherited in a Mendelian fashion and different forms can be manifested in children and adults. RP is characterized by a progressive loss of rod photoreceptors followed by a secondary deterioration of cone photoreceptors, the latter making the disease particularly debilitating [5]. RP is heterogeneous with >4000 identified mutations in >300 genes/loci [6,7]. The link between mutant gene and cell death is not always clear. Some mutant proteins are toxic and directly induce cell death while others perturb metabolic networks that indirectly lead to cell death. The large numbers of different mutations and pathways leading to cell death have hindered the identification of suitable RP treatments. There is currently no cure for RP, although gene therapy, prosthetic implants, and pharmacological agents are all being explored as therapeutic approaches [8,9,10]. The most successful approaches for RP therapy are likely to be gene-independent pharmacological advances with potential to target multiple forms of the disease and can be used for early-stage treatment.

Epigenetic remodeling of gene expression is a powerful new approach to blocking neurodegeneration. Degenerative diseases of the retina, including RP, are good candidates for this strategy because there are both excellent animal models and an available patient population, but no treatments. In our previous published work [11], we demonstrated that inhibition of histone-modifying enzymes LSD1and HDAC1 in a mouse model of RP-altered chromatin structure and led to protection of rods and vision, maintaining expression of rod-specific genes, with reduced cell death, Muller cell gliosis, and inflammation. Because LSD1 and HDAC1 themselves can affect inflammation through the effects of NF-kB activation [12,13,14], these results could not fully resolve the importance of chromatin remodeling.

To address this issue, we have now broadened our studies to test whether multiple classes of inhibitors of chromatin condensation can prevent rod death. We have chosen two enzymes of chromatin organization that act differently to those we have used previously. The first enzyme, EZH2, is a histone lysine methyltransferase that catalyzes trimethylation of histone H3K27. It is a component of the Polycomb Repressive Complex 2 (PRC2) and is critical for both embryonic development and gene silencing. Its methylation activity facilitates both the formation of heterochromatin and heterochromatin remodeling. DZNep (3-deazneplanocin A) and UNC1999 are inhibitors of EZH2 that selectively block formation of H3K27me3. Treatment with DZNep has been reported to slightly delay photoreceptor degeneration in rd1 mice, as well as cause decreased microglial activation in models of ischemic stroke [15,16] and UNC1999 intravitreal injection, partly preventing retina degeneration in two mouse models [17].

The second enzyme, G9A/GLP, is a heterodimeric complex that catalyzes mono- and di-methylation of histone H3K9. The inhibitor UNC0642 is a highly selective inhibitor of this complex, improving cognition and reducing oxidative stress as well as neuroinflammation in a mouse model of Alzheimer’s disease [18]. It also improved survival in the mouse model of Prafer–Willi syndrome [19].

Our results demonstrated that all these inhibitors that likely decondense chromatin decrease retinal degeneration in the mouse model of RP, rd10, but they act through different metabolic pathways. Based on our results, we suggest that these inhibitors of histone-modifying enzymes cause more open and accessible chromatin in rod photoreceptors, which drive the transcription of clusters of neuroprotective genes. Another mechanism that also promotes rod photoreceptor maintenance is an action of inhibitors leading to the suppression of inflammatory processes in microglia and glia.

## 2. Materials and Methods

Reagents. Chemicals were from Fisher Scientific (Pittsburgh, PA, USA). Sodium chloride, saline 0.9% bacteriostatic solution was from APP Pharmaceuticals (Schaumburg, IL, USA). Epigenetic inhibitors GSK2879552, DZNep, and UNC1999 were from Selleckchem.com, URL (accessed on 20 June 2025) (Huston, TX, USA); Romidepsin and UNC0642 were from Sigma (St. Louis, MO, USA).

Antibodies. Anti-GFAP (MAB360) was from Millipore, Temecula, CA, USA, and anti-IBA1 (for the gene Aif1; 019-19741) was from Wako, Richmond, VA, USA. Monoclonal antibodies against rhodopsin (RHO) were described previously [20] and react with an N-terminal sequence shared by many species.

Animals. We used C57Bl/6J (cat#000664), and rd10 B6.CXB1-Pde6brd10/Jrd10 (cat#004297) mice that were from Jackson laboratory (Bar Harbor, ME, USA) and housed in a room with temperature of 25 °C, 30–70% humidity, a 12 h light–dark cycle, and access to rodent chow ad libitum. We were using mice of both sexes in compliance with the National Research Council’s Guide for the Care and Use of Laboratory Animals (8th edition), and experiments with animals were authorized by the Pennsylvania State University College of Medicine Institutional Animal Care and Use Committee (protocol #46993).

To conduct cell sorting of rod photoreceptors expressing GFP from degenerated and GSK-treated retina, we crossed rd10 mouse (B6.CVB1-Pde6brd10/j) with mouse B6.Cg-Tg(Nrl-EGFP)1Asw/J (gift from Dr. A. Swaroop) [21] to establish a mouse line that harbors both homozygous rd10 mutation and homozygous expression of green fluorescence protein in rod photoreceptors. Genotyping was performed according to Jackson lab protocols for rd10 mice (cat#004297) and B6.Cg-Tg(Nrl-EGFP)1Asw/J (cat#021232).

Treatment with inhibitors. All inhibitors were diluted in specific diluent or in 0.9% bacteriostatic sodium chloride (saline). Mice were treated daily with intraperitoneal injections (i.p.) of UNC1999 at 15 mg/kg (diluted in10% DMSO,40% PEG300, 5% Tween-80 and 45% saline), GSK2879552 (GSK, diluted in saline) at 4.2 mg/kg, DZNep at 1–1.5 mg/kg (diluted in saline), Romidepsin at 0.2 mg/kg (diluted in saline), UNC0642 at 7.5 mg/kg (diluted in saline), or saline (diluent) as control.

Tissue collection. We isolated retinas and collected tissue as previously described [11].

FACS. We treated mice from this newly established line from PN9 to PN24 with daily i.p. injections of saline (control) or GSK2879552 at 4.2 mg/kg. Mouse retinas were dissected and dissociate according to a protocol from Solovei’s lab [22]. Cells were sorted on BD FACSDiva 9.0.1 in Flow Cytometry Core at Penn State College of Medicine. RNA was isolated from 2–3 × 10^6^ sorted cells per sample with RNeasy Mini Kit and RNA shredder (Qiagen, Hilden, Germany). RNA integrity number (RIN) was measured using BioAnalyzer (Agilent Technologies, Santa Clara, CA, USA) RNA 6000 Nano Kit to confirm RIN above 9 for each sample.

RNA extraction and cDNA preparation were performed as previously described [11].

RT-PCR. Primers were from Integrated DNA Technologies (IDT) and their sequence information is in Appendix A. qRT-PCR was performed as previously described [11].

RNA-seq were performed as previously described [11,23]. BBDuk (version 38.69) was used to trim adapters and filter low-quality sequences using “qtrim = lr trimq = 10 maq = 10” options and the adapter database provided. Next, alignment of the filtered reads to the mouse reference genome (mouse Ensembl release 67 (GRCm37/NCBIM37/mm9)) was performed using HISAT2 [24] (version 2.2.1), applying --no-mixed and --no-discordant options. Read counts were calculated using HTSeq [25] (version 0.11.2) by supplementing Ensembl gene annotation (release 67: “Mus_musculus.Ensembl.NCBIM37.67.gtf”). Processing of the read counts and differential gene expression were performed with the EdgeR R package [26] (versions 3.42.4 and 4.3.1, respectively). Genes with low counts were filtered using the filterByExpr function. After removal, library sizes were normalized using the calcNormFactors function. The variability in gene expression across all samples was accounted for by estimating the dispersion for each gene using the estimateDisp function. Negative Binomial Generalized Linear Models (NB GLMs) were the fitted-to-the-count data using the glmFit function. Differentially expressed genes were then determined by the likelihood ratio test method (glmLRT function). Significance was defined to be those with q-value < 0.05 calculated by the Benjamini–Hochberg method to control the false discovery rate (FDR), and log2 fold change is greater than 1 or smaller than −1. The pheatmap R package was used for generating heatmaps. Raw counts and differential expression analysis generated during this study are available at GEO Submission GSE295538.

RNA-seq for FACS-sorted rod photoreceptors. RNA-seq libraries were prepared in the Penn State College of Medicine Genome Sciences core (RRID:SCR_021123) using the SMART-Seq mRNA HT LP kit (Takara Bio) as per the manufacturer’s instructions. Briefly, cDNA was synthesized from 1 ng of RNA with recommended PCR cycles of 11 and purified with magnetic beads. Quality and quantity of the cDNA was checked with the BioAnalyzer High Sensitivity DNA Kit (Agilent Technologies). Sequencing libraries were then prepared with 300 pg of the cDNA and indexed with the SMARTer RNA Unique Dual Index Kit. The final product was assessed for its size distribution and concentration using the BioAnalyzer High Sensitivity DNA Kit. The libraries were pooled and sequenced on Illumina NovaSeq 6000 (Illumina) to obtain on average 25 million paired-end 50 bp reads according to the manufacturer’s instructions. Samples were demultiplexed using bclconvert software (version 4.0) (Illlumina). Adaptors were not trimmed during demultiplexing.

Pathway, Gene Ontology, and upstream regulator analysis. The clusterProfiler R package [27] (version 4.8.3) was used to perform GO functional analysis and plot the output. GO terms were reduced using the rrvgo R package (version 1.12.2), and *p*-values used for plotting were the least significant. Ingenuity Pathways Analysis (IPA) was used to identify upstream regulators and significantly enriched canonical pathways with following cutoff of FDR < 0.05 and fold change in gene expression bigger than 2 or smaller than 0.5.

Immunofluorescence Staining. Double-labeling immunohistochemistry was performed as previously described [28,29] with secondary Alexa Fluor-conjugated antibodies diluted 1:800 (Invitrogen, Carlsbad, CA, USA). Primary antibodies were diluted: anti-RHO 1:50, anti-GFAP 1:1000, anti-IBA1 1:450 (Aif1 gene). All slides were counterstained with Hoechst 33,258 (1 mg/mL diluted 1:1000). Images were collected on a Zeiss confocal microscope LSM900.

Statistical Analyses. Results are presented as means ± standard error of the mean (SEM). Unpaired, two-tailed Student’s *t*-test was used for statistical comparison between groups. We used GraphPad Prism software (version 10.4.2) for statistical analysis; *p*-value < 0.05 was considered significant.

## 3. Results

### 3.1. Treatment of rd10 Mouse Model of RP with EZH2 Inhibitors Leads to Retina Neuroprotection

We first treated rd10 mice with EZH2 inhibitor DZNep that was used before in mice [15]. Systemic administration of DZNep to mice could be deleterious, so we first investigated the best timetable for DZNep treatment. The optimal drug administration schedule for animal survival was from PN9 to PN12 each day with 1.5 mg/kg DZNep, and after it, from PN13 to PN24 each second day with 1 mg/kg DZNep. Immunostaining demonstrated preservation of retina outer nuclear layer (ONL) (Figure 1A) and downregulation of GFAP activation after DZNep treatment. After establishing that treatment of rd10 with inhibitor specific for EZH2 led to maintenance of photoreceptors, we studied if this inhibitor affects gene transcription. We carried out RNA-seq on retina samples from rd10 mice treated with DZNep or saline from PN9 to PN24 (Figure 1B). With a cutoff of FRD < 0.05 and a log2 fold change is greater than 1 or smaller than −1, retina samples treated with this inhibitor have 518 genes upregulated and 367 genes downregulated (Figure 1B, Appendix A). Ingenuity Pathways Analysis (IPA) demonstrated that the Visual Phototransduction pathway was the most upregulated pathway, while most downregulated pathways were related to inflammation and immuno-response, integrin cell surface interactions, and interferon gamma signaling (Figure 1C). Cnet plots made from Gene Ontology (GO) functional enrichment analysis showed specific genes that are involved in the upregulated pathways hub of visual perception, detection of light, and sensory perception of light stimulus, as well as the downregulated genes hub of adaptive immune response and antigen processing and presentation (Figure 1D). Top upstream regulators according to IPA were RHO, Immunoglobulin, CRX, TGFB1, TNF, and KDM1a (LSD1). We confirmed RNA-seq results with qRT-PCR for several genes. Rod-specific genes were upregulated and gene markers for inflammation were downregulated under DZNep inhibition in rd10 mice (Appendix AA). GO analysis identified that the 10 most downregulated Molecular Functions (MF) were related to immunity and the extracellular matrix, the 10 most downregulated Cell Components (CC) were connected to the plasma membrane and MHC complexes, and the 10 most downregulated Biological Processes (BPs) were all linked to immunity (Appendix AB). GO analysis identified that the 2 most upregulated Molecular Functions (MF) were retinoid binding and cyclic nucleotide-gated channel activity, the 5 most upregulated Cell Components (CC) included photoreceptor cilium, plasma membrane, and photoreceptor inner segment, and the 10 most upregulated Biological Processes (BPs) were all linked to eye development and function (Appendix AC). Thus, the GO analysis for differently expressed genes (DEGs) between control and DZNep treatment group corroborated the IPA results.

We compared these results with those using a more specific EZH2 inhibitor, UNC1999, to treat the rd10 mouse model. UNC1999 is less toxic for mice and was used in mice before [17]. It was injected at 15 mg/kg from PN9 to PN24. UNC1999 action on rd10 mice was very similar to DZNep, and showed preservation of ONL and downregulation of Muller glia activation (Figure 2A). The number of photoreceptor rows in ONL increased from an average of 5 in rd10 to 10 in retinas of rd10 mice treated with UNC1999 (Figure 2D). We carried out RNA-seq on retina samples from rd10 mice treated with UNC1999 or saline from PN9 to PN24 (Figure 2B). Retina samples treated with this inhibitor have 368 genes upregulated and 519 genes downregulated (Figure 2B, Appendix A). RNA-seq IPA similarly demonstrated preservation of the expression of rod-specific and visual phototransduction pathway genes as well as the downregulation of neuroinflammation genes and complement system pathway (Figure 2C). The top upstream regulators according to IPA were LPS, TNF, RHO, KDM1a (LSD1), IFNG, and Immunoglobulin. We confirmed the RNA-seq results with qRT-PCR for selected genes. Rod-specific genes were upregulated and genes specific for inflammation were downregulated under UNC1999 inhibition in rd10 mice (Appendix AA). GO analysis identified that the 10 most downregulated Molecular Functions (MF) were related to immunity and transmembrane receptor binding, the 10 most downregulated Cell Components (CC) were connected to plasma membrane and MHC complexes, and the 10 most downregulated Biological Processes (BPs) were all linked to immunity (Appendix AB). GO analysis identified that the 2 most upregulated Molecular Functions (MF) were anion transporter activity and extracellular matrix structure, the 10 most upregulated Cell Components (CC) included photoreceptor cilium, plasma membrane, and photoreceptor inner segment, and the 10 most upregulated Biological Processes (BPs) were all linked to eye development and function (Appendix AC). Thus, the GO analysis for differently expressed genes (DEGs) between control and UNC1999 treatment group corroborated the IPA results.

### 3.2. Treatment of rd10 Mouse Model of RP with G9A/GLP Inhibitor Leads to Partial Retina Neuroprotection

During rod photoreceptor maturation, their nuclei undergo dramatic transformation, so that adult retina rods have only one big heterochromatic focus that occupies much of the nuclei volume. The epigenetic mark H3K9me2 is associated with facultative heterochromatin, so we used UNC0642, inhibitor of the histone modification complex G9A/GLP that catalyzes mono- and di-methylation of histone H3K9, to study if relaxing heterochromatin in rd10 mice in such a way will help the rod photoreceptor to survive. This inhibitor was used before in mice [18]. Treatment with the UNC0642 inhibitor partially preserved the rod photoreceptor, retina layer structure, and lower gliosis (Figure 3A). The number of photoreceptor rows in ONL increased from an average of 4 for rd10 to 7–8 in retinas of rd10 mice treated with UNC0642 (Figure 3C). The lower efficacy of UNC0642 was probably due to low solubility in aqueous solutions and known low permeability across the blood–brain barrier [30]. We carried out RNA-seq analysis on retina samples from rd10 mice treated with UNC0642 or saline from PN9 till PN24 (Figure 3B). Retina samples treated with this inhibitor have only 44 genes upregulated and 28 genes downregulated (Figure 3B, Appendix A). IPA demonstrated that this limited set of genes participate mostly in upregulating stress response, including ERK5 signaling, NGF-stimulated transcription, NRF2-medited oxidative response, and downregulation of EIF2 and DHCR24 signaling responses (Figure 3D). EIF2 signaling is connected with upregulation of transcription and DHCR24 signaling—with cholesterol synthesis.

### 3.3. RNA-Seq of Retina of rd10 Mice Treated with HDAC1 Inhibitor

In our previous paper [11], we demonstrated that just as inhibitors of LSD1 block rod photoreceptor death in rd10 mice, an inhibitor of histone deacetylase HDAC1, Romidepsin, has similar effect (see [11]—Figure 1). While under Romidepsin inhibition, rod-specific genes were upregulated to a much lesser extent as compared with under GSK treatment (see [11]—Figure 5A), downregulation of immune- and inflammatory-specific genes was robust (see [11]—Figure 5F). For proper comparison between different epigenetic inhibitors, we carried out RNA-seq analysis of retina samples from rd10 mice treated with Romidepsin or saline from PN9 to PN24 (Figure 4B). Retina treated with this inhibitor have 511 genes upregulated and 402 genes downregulated (Figure 4B, Appendix A). Rod-specific genes such as Rho, Aipl1, Pde6b, Ne2e3, Guca1a, Pdc, and Ptp4a3 were upregulated, while cone-specific genes Gnat2, Pde6c, and Opn1mw were downregulated. But mostly genes with altered expressions belong to immune and inflammation response pathways. IPA demonstrated association of HDAC inhibition with inflammation and extracellular matrix organization pathways (Figure 4A). The top upstream regulators according to IPA were LPS, b-estradiol, 1L1b, immunoglobulin, IFNG, and IL6. Cnet plots made from Gene Ontology (GO) functional enrichment analysis showed specific DEGs that are involved in pathways hubs of response to bacteria, immune effector process, regulation of defense response, and negative regulation of viral process (Figure 4C). GO analysis identified that the 10 most downregulated Molecular Functions (MF) were related to receptor activity, extracellular matrix, and binding to it; the 5 most downregulated Cell Component (CC) were connected to extracellular matrix; and MHC complexes and the 10 most downregulated Biological Process (BP) were all linked to immunity, defense response, and NF-kB signaling (Appendix A). GO analysis identified that the 2 most upregulated Molecular Functions (MF) were DNA-binding/RNA polymerase transcription and cyclin-dependent inhibitor activity, and the 10 most upregulated Biological Processes (BPs) were all linked to cell differentiation and proliferation (Appendix A). Thus, GO analysis for DEGs between control and Romidepsin treatment groups corroborated with IPA.

### 3.4. Comparison of Action of Different Inhibitors of Chromatin Compaction

All of the inhibitors of chromatin compaction: Romidepsin for HDAC, as well as DZNep and UNC1999 for EZH2 and G9A/GLP inhibitor UNC0642, preserved rod photoreceptors from degeneration in the mouse model of RP, rd10. To find common biological pathways and mechanisms to fight retina degeneration, we compare mRNA-seq data for samples presented in this manuscript and related it to our previous mRNA-seq data for GSK2879552-treated retinas [11]. The overall comparison revealed that one group of epigenetic inhibitors—DZNep, UNC1999, and GSK—have much more common upregulated genes than another group, UNC0642 and Romidepsin (Figure 5A,B).

Approximately half of the common upregulated DEGs for GSK, DZNep, and UNC1999 treatments belong to visual phototransduction and visual cycle pathways and included such retina developmental TFs such as Nrl and Cazs1 (Figure 5D). In addition, there were several common upregulated pathways that are important in supporting photoreceptor survival: Ca^2+^ metabolism, potassium ion transport, and positive regulation of Wnt signaling pathway (Figure 5D). Common downregulated DEGs for GSK, DZNep, and UNC1999 treatments belong to cellular immune response pathways, antigen processing and presentation, and lymphocyte differentiation (Figure 5E).

Treatments with Romidepsin and UNC0642 gave different results from the other three inhibitors. Fewer genes belonging to phototransduction and visual cycles were robustly upregulated in this group, though a number of others were upregulated to a lesser extent. Several of upregulated genes are members of immediate–early response genes (IER)—Fosb, Fosl1, Gadd45a, Gadd45b, Myc, and Junb. These genes participate in NGF-stimulated transcription, transcriptional activation, and NRF2-mediated oxidative stress responses. Common upregulated processes for these inhibitors include cell differentiation, cellular response to TNFa, and cytoskeleton organization (Figure 5F). Common downregulated DEGs for Romidepsin and UNC0642 treatments belong to pathways associated with extracellular matrix, neuroinflammation signaling, and cholesterol and programmed cell death (Figure 5G).

### 3.5. Changes in Gene Expression of Isolated Rod Photoreceptor in rd10 Mice Treated with GSK2879552

Whole-retina RNA-seq gives a complex picture of gene expression changes, especially for low-expressing transcript, as different types of cells could demonstrate opposite changes for the same gene. Even for the largest cell population such as rod photoreceptors, changes in low-expressed but important rod genes could be skewed because of altered expression of the same gene in other cell types. To identify changes specific to rod photoreceptors, we crossed rd10 mice with mice that expressed EGFP in rods under the Nrl promoter [21] and established a mouse line that is homozygous for both the rd10 mutation and expression of green fluorescence protein in rod photoreceptors. The treatment of this line with GSK had the same protective effect as treatment with rd10. Pure rod photoreceptors were FACS-sorted from dissociated retinas of saline- and GSK-treated animals and rod-specific RNA was subjected to RNA-seq. Sorted rod photoreceptors had 25 genes that were downregulated and 46 genes that were upregulated (Figure 6A, Appendix A). Among the downregulated genes were Edn2, a known marker of retina degeneration, Gadd45b, a stress response gene, and Igrm1, a gene that positively regulates macroautophagy. Two cone-specific genes, Pde6h and Gnb3, were also downregulated. The majority of upregulated genes had low expression at PN24 (according to [31]; https://viz.stjude.cloud/dyer-lab/visualization/the-4-dimensional-nucleome-of-murine-rod-photoreceptors~72; (accessed on 12 June 2025)), while 7 genes are located in retina developmental enhancer, cis-regulatory DNA sequences that orchestrate gene expression during retina development. To find out which part of chromatin a gene resides in, we used the published data of Aldiri et al. [31] at https://viz.stjude.cloud/dyer-lab/visualization/the-4-dimensional-nucleome-of-murine-rod-photoreceptors~72, (accessed on 12 June 2025). The authors predicted chromatin state of the genome loci of particular genes (euchromatin or heterochromatin) at E14.5 or PN21 developmental stages. As we completed our experiments at PN24, we used the PN21 chromatin predictions from this source. One-third of upregulated genes resided in heterochromatin, again emphasizing that treatment with epigenetic inhibitors decondensed heterochromatin, leading to re-expression of these genes. Two examples of genes residing in rod photoreceptor heterochromatin and having no retina expression at this age but became upregulated in retinas treated with GSK: *Kit*, important for regeneration and homeostasis; and *Sstr*, important for photoreceptor development and it inhibits calcium entry by suppressing voltage-dependent calcium channels. According to IPA (Figure 6B), most upregulated pathways by GSK in the sorted rods were related to collagen metabolism, G-protein-coupled receptor signaling, RAF/Map kinase cascade, and CREB signaling in neurons.

We compared our RNA-seq data on sorted rods (rd10 vs. rd10 + GSK) and data of rod photoreceptors from scRNA-seq of rd10 mice at PN21 (cluster C1 vs. clusters C2, C3) [32]. We used the same relaxed cutoff for our data that was used for the scRNA-seq: *p* < 0.05, log FC > 0.25. There were 41 genes in common between the two sets. Most of these genes were upregulated during degeneration of rods and downregulated to near-normal levels under GSK treatment. These included genes regulating apoptosis or stress response genes *Cd47*, *E2f6*, *Gadd45b*, *Edn2*, and *Wwtr1*. Interestingly, there were only 3 genes that were downregulated during degeneration but were upregulated under GSK treatment. They are as follows: (1). *Hk2*, hexokinase, which functions in the negative regulation of mitochondrial membrane permeability, negative regulation of reactive oxygen species metabolism, anti-apoptotic, and photoreceptor protection; (2). *Rabgef1*, with functions in ubiquitin protein ligase activity, dendritic transport, with the loss of it causing aberrant morphogenesis and altered autophagy in photoreceptors leading to retinal degeneration; (3). *Selenbp1*—selenium binding protein 1—that catalyzes the oxidation of methanethiol and plays a role in intra-Golgi protein transport. In addition to the effects of GSK in sorted rod photoreceptors, all three were upregulated in rd10 treated with DZNep, Romidepsin, and UNC1999.

## 4. Discussion

In our previous study, we found that epigenetic inhibitors GSK2879552 and Romidepsin block rod degeneration, preserve vision and transcription of rod photoreceptor genes, and downregulate transcripts involved in cell death, gliosis, and inflammation in the mouse model of Retinitis Pigmentosa (RP), rd10 [11]. GSK2879552 and Romidepsin inhibit different enzymes, LSD1 and HDACs, respectively, but these enzymes have common characteristics—they condense chromatin inside the nucleus of the cell and sometimes work in the same protein complex.

To better understand whether chromatin decondensation is a gene-independent way to prevent retina degeneration, we have investigated whether other inhibitors of chromatin condensation can prevent photoreceptor degeneration in the rd10 mouse model. We tested inhibitors for G9A/GLP, which catalyzes methylation of H3K9, and for EZH2, which catalyzes trimethylation of H3K27, and compared them to the actions of LSD1 and HDAC inhibitors. We showed that all these inhibitors that likely decondense chromatin are preserving retina from degeneration in mouse model of RP, rd10. The combined results provide robust support for our hypothesis that inhibition of chromatin condensation can be sufficient to prevent rod death in rd10 mice.

It is clear that the less compact chromatin allows major changes in gene expression. We observed strong upregulation of rod genes involved in visual transduction in the whole-retina RNA-seq, but not in the isolated rods. This strongly suggests that the observed changes are a consequence of changing numbers of rods. A question that arises is whether all of the inhibitors lead to up- or downregulation of a few key genes, or whether neuroprotection is a broad phenomenon that can be achieved in multiple ways. To address this, we carried out RNA-seq studies for each inhibitor treatment and compared DEGs and molecular pathways changes for all epigenetic inhibitors. Our previous hypothesis for inhibitors action proposed two mechanisms: (1) epigenetic modifiers cause more open and accessible chromatin in rods, which drive the transcription of clusters of neuroprotective genes and surviving rods; (2). suppression of inflammation in microglia and Muller glia. These two processes could be separated or work together as positive feedback loop where normal photoreceptors and inhibitor-treated photoreceptors do not produce any signaling molecules to activate inflammation.

In addition to their effects on visual phototransduction pathways and overall rod preservation, inhibitors of LSD1 and EZH2 caused an upregulation of genes from neuroprotective Ca^2+^ and Wnt pathways. Additionally, these modifiers downregulate inflammation pathways, such as Complement System, Leukocyte Extravasation Signaling, Natural Killer Cell Signaling, Neuroinflammation Signaling Pathway, Th2 Pathway, and Toll-like Receptor Signaling. Since the changes in inflammation pathways were not detected in isolated treated rods, they probably arose in either microglia or Muller cells, both of which participate in inflammatory responses in the retina. At present we cannot distinguish whether the inhibitors are acting directly on microglia and Muller cells or whether they induce signals in rods that, in turn, affect the glial cells.

Some of the inhibitors may also have direct effects on inflammatory pathways. The LSD1 enzyme [12] participates in NF-kb activation and the same has been suggested for EZH2 [33,34]. Our data demonstrate that KDM1a (LSD1) is the 4th top upstream regulator for UNC1999 and 6th for DZNep, indicating that these epigenetic modifiers inhibits or activates common pathways. Such dual actions of some of the inhibitors could potentially make them more potent protective agents.

Both inhibitors of HDACs and GLP/G9a demonstrate upregulation of NGF-stimulated transcription but they mostly downregulate inflammation by inhibiting IL-12 signaling and production in macrophages, interferon alpha/beta signaling, interferon gamma signaling, Neuroinflammation Signaling Pathways, and Phagosome Formation. This suggests that inhibitors of HDACs and GLP/G9a mostly work through the second mechanism proposed above—suppression of inflammation in microglia and Muller glia. HDAC inhibitors promote photoreceptor survival in several mouse models of RP and AMD [35,36,37,38], and there are numerous studies showing that HDAC inhibitors ameliorate inflammation [13,14,39,40].

It was shown that rearing rd10 mice in the dark preserved photoreceptors from degeneration through inactivation of rhodopsin signaling [41], but exposure to room light is followed by rapid photoreceptor degeneration [42]. These experiments suggest that blocking rod function under conditions of excess of phototransduction/light could partially rescue retinal degeneration. For inhibitors of HDAC1 (Romidepsin) and G9a (UNC0642), our data showed almost no changes for rod-specific genes when, paradoxically, the number of rod photoreceptor is much bigger than in controls (Figure 3C), so it is possible that we preserved the retina by blocking rod function. It will be important in the future to perform scRNA-seq and ERG measurements using mouse retinas to better understand mechanisms of action.

Retina-specific knockout or knockdown of histone modification enzymes is usually performed at embryonic stages, before rod photoreceptors properly differentiate and exit the cell cycle, which can lead to poor retinal development or degeneration [43,44]. In our experiments, we treated rd10 mice with epigenetic enzyme inhibitors after PN9, when rods have exited the cell cycle and genes that not expressed in rods begin accumulating the H3K27me3 mark and shut down [4,31]. The time of manipulation of epigenetic enzymes is very important, as a lot of them work differently at different points in development [45].

In summary, our findings indicate that multiple epigenetic modifiers can block degeneration of rod photoreceptors in a model of RP. They can achieve this by altering expression of protective and injurious genes in rods as well as modifying the expression of inflammatory pathways in other retinal cells. These multiple mechanisms suggest that epigenetic modifiers are ideal agents to allow cells to reset homeostasis and counteract deleterious effects of mutations or other disease-causing stimuli. Such modifiers may prove to be valuable gene-independent therapeutics in targeting a range of multifactorial diseases in the retina, such as macular degeneration and diabetic retinopathy, as well as elsewhere in the nervous system.

## Figures and Tables

**Figure 1 cells-14-00961-f001:**
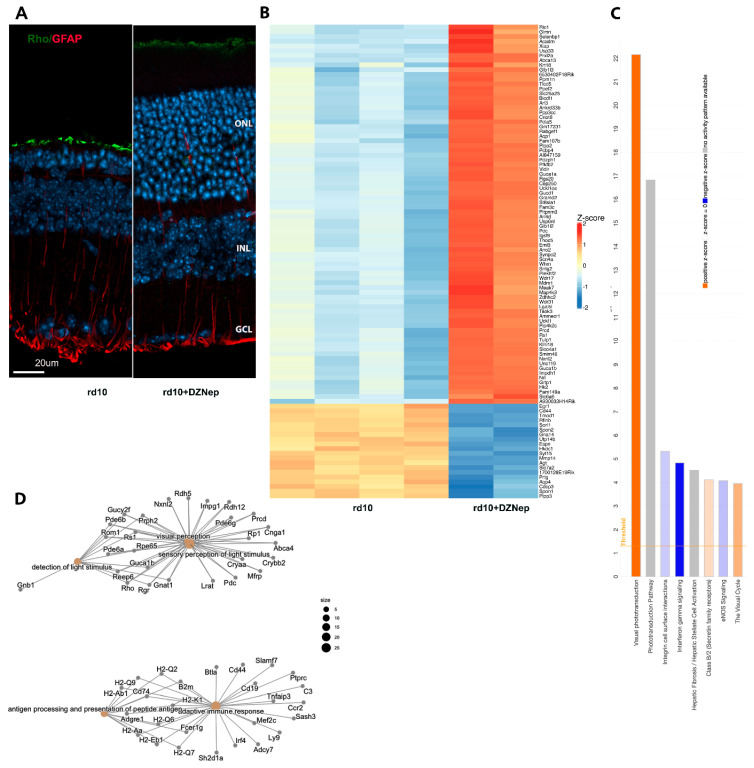
Treatment of rd10 mice with EZH2 inhibitor DZNep leads to retina neuroprotection. (**A**). Immunofluorescence (IF) images of retina sections from PN24 mice injected with DZNep or only saline, stained with anti-RHO (green), anti-GFAP (red) antibodies. Nuclei were stained with Hoechst33258. (**B**). RNA-seq analysis of differentially expressed genes (DEGs) under DZNep treatment. Heatmap of 100 most changed DEGs (FDR < 0.05; log2 fold change is greater than 1 or smaller than −1). (**C**). Top IPA canonical pathways for DEGs (FDR < 0.05; log2 fold change is greater than 1 or smaller than −1). (**D**). Category–gene network plot depicting DEGs.

**Figure 2 cells-14-00961-f002:**
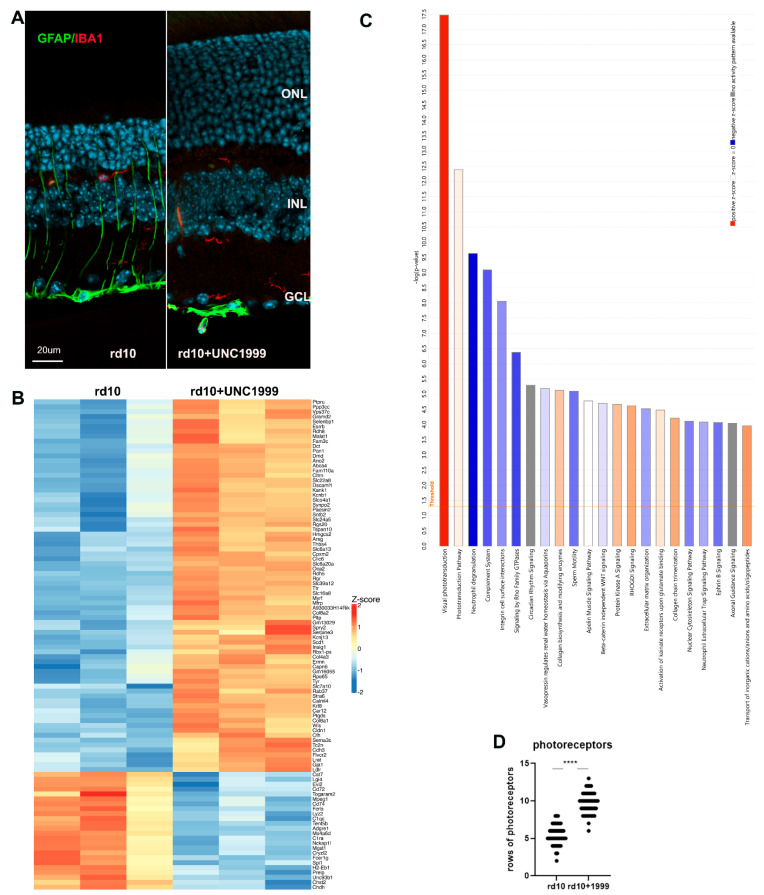
Treatment of rd10 mice with EZH2 inhibitor UNC1999 leads to retina preservation. (**A**). IF images of retina sections from PN24 rd10 mice injected with UNC1999 or only diluent stained with anti-GFAP (green), anti-IBA1 (red) antibodies. Nuclei were stained with Hoechst33258. (**B**). RNA-seq analysis of DEGs under UNC1999 treatment. Heatmap of 100 most changed DEGs (FDR < 0.05; log2 fold change is greater than 1 or smaller than −1). (**C**). Top IPA canonical pathways for DEGs (FDR < 0.05; log2 fold change is greater than 1 or smaller than −1). (**D**). Rod row counts in central retina of rd10 PN24 injected with UNC1999 or only diluent for 3 biological and 3 technical replicas (**** *p* < 0.0001).

**Figure 3 cells-14-00961-f003:**
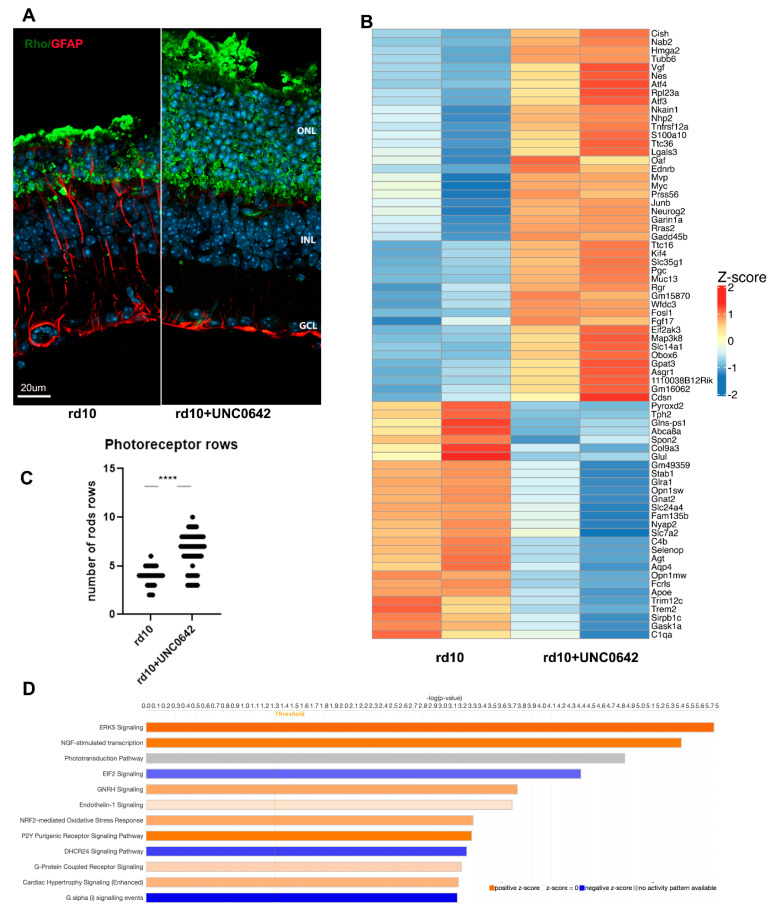
Treatment of rd10 mice with G9a/GLP inhibitor UNC0642 leads to partial retina protection. (**A**). IF images of retina sections from PN24 rd10 mice injected with UNC0642 or only saline stained with anti-RHO (green), anti-GFAP (red) antibodies. Nuclei were stained with Hoechst33258. (**B**). RNA-seq analysis of DEGs under UNC0642 treatment. Heatmap of 100 most changed DEGs (FDR < 0.05; log2 fold change is greater than 1 or smaller than −1 (**C**). Rod row counts in central retina of rd10 PN24 injected with UNC0642 or only saline for 2–3 biological and 3 technical replicas (**** *p* < 0.0001). (**D**). Top IPA canonical pathways for DEGs (FDR < 0.05; log2 fold change is greater than 1 or smaller than −1).

**Figure 4 cells-14-00961-f004:**
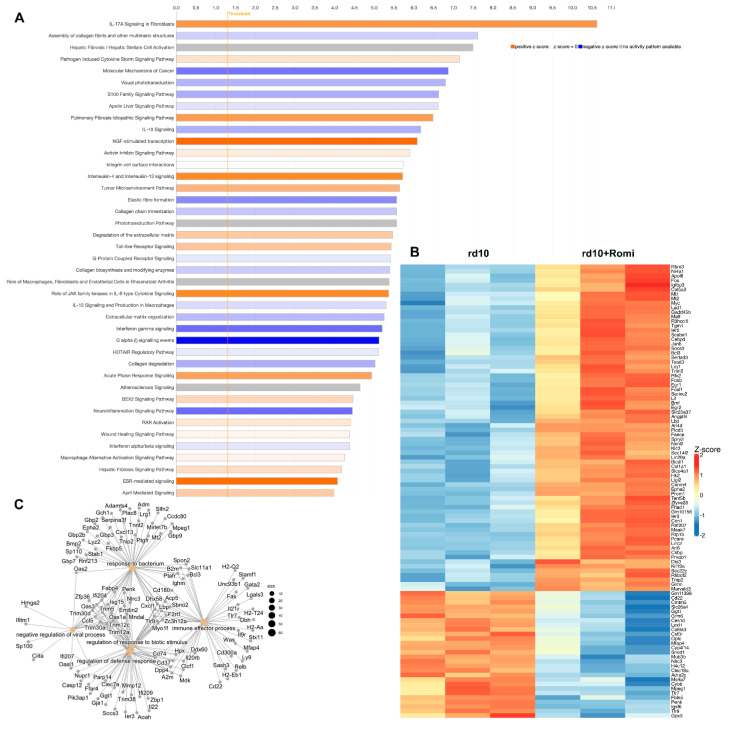
RNA-seq analysis of DEGs in rd10 mice under HDAC inhibition with Romidepsin. (**A**). Top IPA canonical pathways for DEGs (FDR < 0.05; log2 fold change is greater than 1 or smaller than −1). (**B**). Heatmap of 100 most changed DEGs (FDR < 0.05; log2 fold change is greater than 1 or smaller than −1). (**C**). Network plot of enrichment functions by cnet plots for DEG.

**Figure 5 cells-14-00961-f005:**
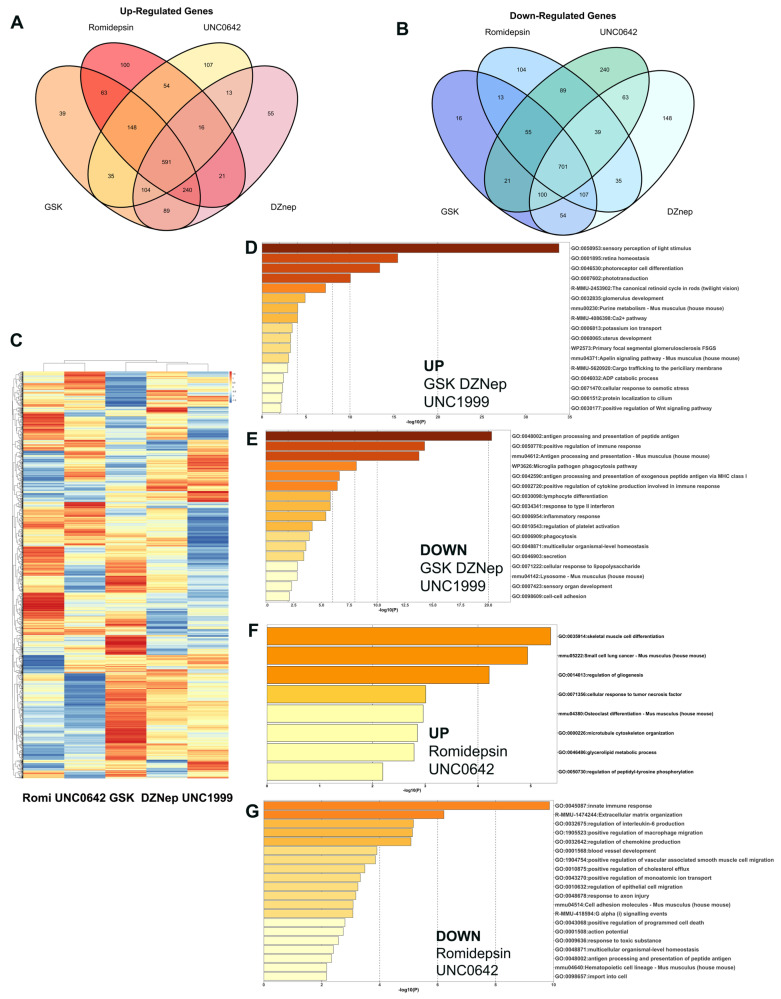
Comparison of action of different inhibitors of chromatin compaction on gene expression in rd10 mouse model of Retinitis Pigmentosa. (**A**). Venn diagram representing overlapping upregulated DEGs identified by RNA-seq analysis among GSK-, DZNep-, UBC0642-, and Romidepsin-treated groups. (**B**). Venn diagram representing overlapping downregulated DEGs identified by RNA-seq analysis among GSK-, DZNep-, UBC0642-, and Romidepsin-treated groups. (**C**). Heatmap for comparison of DEG identified by RNA-seq analysis among GSK-, DZNep-, UBC0642-, UNC1999-, and Romidepsin-treated groups. (**D**–**G**). Bar plots representing enriched ontology clusters for common upregulated (**D**) and downregulated (**E**) DEGs identified by RNA-seq analysis among GSK-, DZNep-, and UNC1999-treated groups; and for common upregulated (**F**) and downregulated (**G**) DEGs identified by RNA-seq analysis among Romidepsin- and UNC0642-treated groups.

**Figure 6 cells-14-00961-f006:**
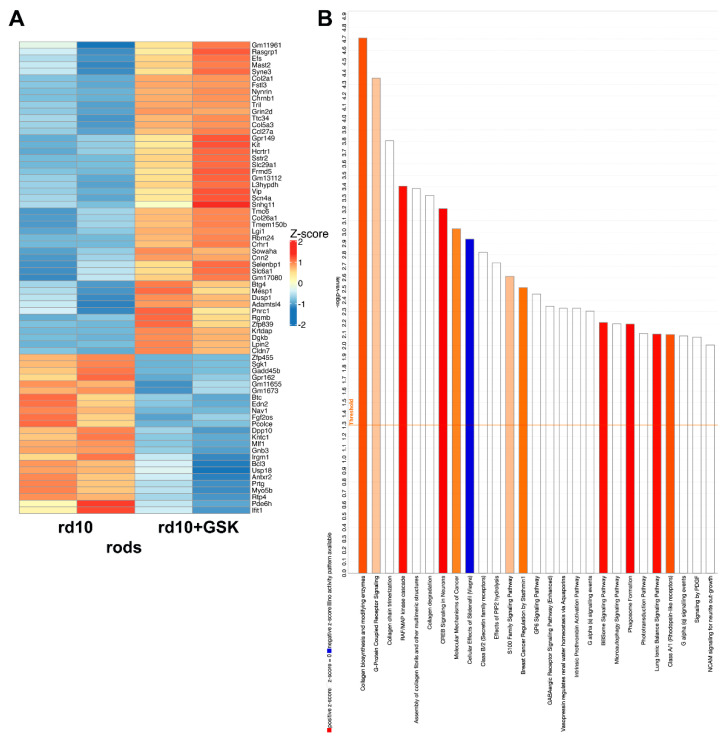
RNA-seq analysis of DEGs of FACS-sorted rods from rd10 mice injected with GSK2879552 vs. only saline rd10 (control). (**A**). Heatmap of 100 most changed DEGs (FDR < 0.05; log2 fold change is greater than 1 or smaller than −1). (**B**). Top IPA canonical pathways for DEGs (FDR < 0.05; log2 fold change is greater than 1 or smaller than −1).

## Data Availability

Raw read counts and differential expression analysis from this study are available at GEO Submission GSE295538.

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
