# Peer review of "Epigenetic Modifiers to Treat Retinal Degenerative Diseases"

_cells, 2025, doi:10.3390/cells14130961_

Round 1

Reviewer 1 Report

Comments and Suggestions for Authors

Photoreceptor degeneration is a tremendously widespread problem that affects millions of people, and novel treatments are badly needed. Here, Popova et al. address the idea that pharmacological inhibition of heterochromatic processes can be neuroprotective to rods. They use the Rd10 model of retinitis pigmentosa, which has been widely studied, and harbors a mutation in the Pde6b gene. As PDE6B is also mutated in human retinitis pigmentosa, this is an appropriate approach. The study builds off of a compelling line of investigation by the Barnstable laboratory that characterized epigenetic processes in rods both fundamentally, and translationally. In particular, the authors recently showed that pharmacological inhibition of histone deacetylases and LSD1 (Kdm1a) leads to dramatic rescue of rod photoreceptor survival in the Rd10 retina.

In the present manuscript, the authors build off of this discovery in two ways. First, they extend the analysis to characterize the effects of additional drugs that affect heterochromatic processes. Second, they perform transcriptomic profiling, both on bulk retinal samples, and on sorted rods. These data are carefully analyzed in order to draw out a variety of interesting observations that can help to explain the neuroprotective effects. Interestingly, the paper finds that the drugs appear to have different transcriptomic effects that could potentially promote rod survival via different mechanisms. These observations constitute a great contribution to the field, and I expect that the paper will be of interest to a variety of other research groups, since the paper shows compelling preclinical effects, provides mechanistic insight, and is focused on the interface between epigenetics and neuroprotection, for which there is great interest. 

I think that this is a strong and well-written manuscript that is built on a foundation of excellent data. The authors were careful in their interpretations. They supplemented their whole-retina transcriptional profiling with data from sorted rod photoreceptors in order to draw conclusions about transcriptome changes in the retina as a whole, as well as the effects that were specifically confined to rods. I have only minor comments.

Minor comments:

Probably the main question/criticism that I have, is the question of whether the epigenetic inhibitors work by blocking rod function. Epigenetic inhibitors have been mainly studied in Rd1, Rd10, and light damage models. In these models, rod cell death is driven by excessive or uncontrolled phototransduction. If the epigenetic inhibitors function by simply blocking or blunting rods from being able to phototransduce light, they would be naturally resistant to degeneration. Along these lines, genetic manuipulation of heterochromatic processes (e.g. Bmi1 mutants, Ezh2 mutants, G9a/GLP mutants) can lead to photoreceptor degeneration in wild-type (i.e. non RP) retinas, and ERGs performed in some of these lines revealed loss of the A-wave (e.g. Barabino et al., Yan et al. Scientific Reports) prior to degeneration. The authors' prior paper in J. Neurosci did address vision, but via the optomotor reflex, which I believe likely reflects cone function more than rod function. ERGs are probably a better way to address this question. I feel that these ideas should probably be addressed in the discussion, if only because sustained treatment with these inhibitors might actually compromise vision, or even drive degeneration.

The other criticism that I think might be important, is that some of the figure elements are too small. There are labels in some of the heatmaps that cannot be read. The font sizes for the GO terms are almost too small to read in Fig. 6. The scale legends are too small in most of the figures (e.g. Fig. 4A, 5C, 6). Figures can definitely be revised and polished in order to present the data more clearly.

Abstract: "All the inhibitors decondense chromatin"

I agree that this is probably true. However, the paper does not actually demonstrate this directly. Therefore, it is probably safer to say that the inhibitors 'should' or 'likely' lead to chromatin decondensation (or just remove the statement).

First paragraph: "Mature rod photoreceptors, like most neurons, belong to the second group of cells and have a uniquely closed chromatin organizational structure [1-3]."

Strictly speaking, this uniquely closed genome architecture has only been demonstrated for mouse rods. The current consensus in the literature would predict that the rods of diurnal species (including humans) would not exhibit this closed architecture. I think that there is also a tension in the introduction, in that the text leaves ambiguous the distinction between gene-poor constitutive heterochromatin, and facultative heterochromatin that is suppressed by H3K27me3. Most heterochromatinized genes are suppressed by the latter (even in murine rods), but this type of heterochromatin is much more dynamic, and much less compacted. While I think that the introductory paragraph is very well written and enjoyed the portrayal, I actually disagree with the content in the two respects listed above. I think the authors should have license to write the introduction as they see fit, but I do think it could be adjusted to take into consideration these nuances.

Line 224: "Unpaired, one tail Student’s t-test (two-tailed, unpaired) was used to evaluated statistical significance..."

One-tailed or two tailed? Please reconcile.

Line 234: "Immunostaining demonstrated preservation of retina outer nuclear layer of rods"

However, the ONL also contains cones (as well as Muller processes).

Line 432: "One third of upregulated genes resided in heterochromatin".

How do you know this? Solovei et al. (2009) showed previously that genes that are not expressed in rods can still be localized to euchromatic regions of the nucleus. I think this is an overly strong conclusion that warrants some qualification.

Reference 5 should be fixed. The cited reference is a 'News & Views' article that refers to the following paper: PMID: 31964843. I think that this latter citation should be used instead.

Comments on the Quality of English Language

The paper is very well written, but there were a few trivial typographical errors.

Author Response

Response for reviewers’ comments.

Dear Editor and Reviewers,

We appreciate the careful review of our manuscript and were encouraged by the positive comments and interest in our work.  We answered questions raised by the reviewers point by point below. Additionally, we changed some wording to eliminate repetition/similarity between this manuscript and our previous paper (Popova, 2021). All edits are highlighted in red in manuscript.

  1. Comments 1. Probably the main question/criticism that I have, is the question of whether the epigenetic inhibitors work by blocking rod function. Epigenetic inhibitors have been mainly studied in Rd1, Rd10, and light damage models. In these models, rod cell death is driven by excessive or uncontrolled phototransduction. If the epigenetic inhibitors function by simply blocking or blunting rods from being able to phototransduce light, they would be naturally resistant to degeneration. Along these lines, genetic manuipulation of heterochromatic processes (e.g. Bmi1 mutants, Ezh2 mutants, G9a/GLP mutants) can lead to photoreceptor degeneration in wild-type (i.e. non RP) retinas, and ERGs performed in some of these lines revealed loss of the A-wave (e.g. Barabino et al., Yan et al. Scientific Reports) prior to degeneration. The authors' prior paper in J. Neurosci did address vision, but via the optomotor reflex, which I believe likely reflects cone function more than rod function. ERGs are probably a better way to address this question. I feel that these ideas should probably be addressed in the discussion, if only because sustained treatment with these inhibitors might actually compromise vision, or even drive degeneration.

Response 1. The Reviewer raised a very interesting point about blocking rod function in conditions of excess of phototransduction to partially fight retina degeneration. In such case there will be no night vision (nonfunctional rod), but cones will survive. Our data for inhibitors of LSD1 (GSK) and inhibitors of EZH2 (DZNep, UNC1999) demonstrate upregulation of rod-specific transcript in bulk mRNA-seq and no changes for rod genes in sorted rods RNA-seq. Yan et al, 2016 actually showed upregulation of rod-specific genes (bulk microarray) after KO of EZH2. They knocked out EZH2 at E13.5, whereas we started injections of inhibitors at PN9. The time for manipulation of epigenetic enzymes is very important, as many of them work differently at different developmental stages (Kerenyj MA et al Elife, 2016). For inhibitors of HDAC1 (Romidepsin) and G9a (UNC0642), our data show almost no changes for rod-specific genes (bulk mRNA-seq), so it could be that we are partially blocking rod function. It is possible that even if we preserve retina at PN24, we might somehow compromise vision. We agree with the reviewer that it will be important in the future to perform ERG to understand specific mechanisms of action of these epigenetic inhibitors.

We added this to the Discussion from line 480:

It was shown that rearing rd10 mice in the dark preserved photoreceptors from degeneration through inactivation of rhodopsin signaling [41], but exposure to room-light is followed by rapid photoreceptor degeneration [42].  These experiments suggest, that blocking rod function in conditions of excess of phototransduction/ light could partially rescue retinal degeneration. For inhibitors of HDAC1 (Romidepsin) and G9a (UNC0642), our data showed almost no changes for rod-specific genes when paradoxically the number of rod photoreceptor is much bigger than in controls (Fig.3C), so it is possible that we preserved the retina by blocking rod function. It will be important in the future to perform scRNA-seq and ERG measurements using mouse retinas to better understand mechanisms of action.

Retina-specific knockout or knockdown of histone modification enzymes is usually done at embryonic stages, before rod photoreceptors properly differentiate and exit the cell cycle, which can lead to poor retinal development or degeneration [43, 44]. In our experiments we treated rd10 mice with epigenetic enzyme inhibitors after PN9, when rods have exited the cell cycle and genes that not expressed in rods begin accumulating the H3K27me3 mark and shutting down [4, 31]. The time of manipulation of epigenetic enzymes is very important, as a lot of them work differently at different point in development [45].

2.Comments 2. The other criticism that I think might be important, is that some of the figure elements are too small. There are labels in some of the heatmaps that cannot be read. The font sizes for the GO terms are almost too small to read in Fig. 6. The scale legends are too small in most of the figures (e.g. Fig. 4A, 5C, 6). Figures can definitely be revised and polished in order to present the data more clearly.

Response 2. We tried to improve the quality of the figures and included in heatmaps only 100 most changed genes. Additionally, we supplemented heatmaps, with a list of genes that show significantly expression changes (FDR<0.05) for each inhibitor/condition in Supplement table 2.

3.Commnets 3. Abstract: "All the inhibitors decondense chromatin. "I agree that this is probably true. However, the paper does not actually demonstrate this directly. Therefore, it is probably safer to say that the inhibitors 'should' or 'likely' lead to chromatin decondensation (or just remove the statement).

Response 3. We agree with the reviewer that we have demonstrated decondensation only for GSK in our previous work and not for the rest of inhibitors, so we used word “likely” in the Abstract.

  1. Comments 4. First paragraph: "Mature rod photoreceptors, like most neurons, belong to the second group of cells and have a uniquely closed chromatin organizational structure [1-3]."Strictly speaking, this uniquely closed genome architecture has only been demonstrated for mouse rods. The current consensus in the literature would predict that the rods of diurnal species (including humans) would not exhibit this closed architecture. I think that there is also a tension in the introduction, in that the text leaves ambiguous the distinction between gene-poor constitutive heterochromatin, and facultative heterochromatin that is suppressed by H3K27me3. Most heterochromatinized genes are suppressed by the latter (even in murine rods), but this type of heterochromatin is much more dynamic, and much less compacted. While I think that the introductory paragraph is very well written and enjoyed the portrayal, I actually disagree with the content in the two respects listed above. I think the authors should have license to write the introduction as they see fit, but I do think it could be adjusted to take into consideration these nuances.

Response 4. We agree with the reviewer and added a paragraph in the Introduction after line 49 to address these issues:

In murine rod photoreceptors chromatin forms layers with distinctive patterns of posttranslational histone modifications. Constitutive heterochromatin is found in center of the nuclei marked by H3K9me3 and H4K20me3. During rod maturation, a large-scale gene repression occurs, creating layer of facultative heterochromatin around central aria marked by H3K27me3 and H3K9me2. The euchromatin layer, containing active genes marked by histone acetylation and H3K4me2/3, is located at the nuclear periphery [1, 2, 4]

5.Comments 5. Line 224: "Unpaired, one tail Student’s t-test (two-tailed, unpaired) was used to evaluated statistical significance..."

Response 5. We used two-tailed test and corrected this mistake in the text.

6.Comments 6. Line 234: "Immunostaining demonstrated preservation of retina outer nuclear layer of rods" However, the ONL also contains cones (as well as Muller processes).

Response 6. We removed “rods” from this sentence.

7.Comments 7.  Line 432: "One third of upregulated genes resided in heterochromatin". How do you know this? Solovei et al. (2009) showed previously that genes that are not expressed in rods can still be localized to euchromatic regions of the nucleus. I think this is an overly strong conclusion that warrants some qualification.

Response 7. To find out in which part of chromatin a gene resides we used the published data of Aldiri et al. (2017). This data are present at https://viz.stjude.cloud/dyer-lab/visualization/the-4-dimensional-nucleome-of-murine-rod-photoreceptors~72. Here the authors predicted chromatin state of the genome where particular genes could be found (euchromatin or heterochromatin) at E14.5 or PN21 developmental stages. As we completed our experiments at PN24, we used the PN21 chromatin predictions from this source.  We have incorporated this more detailed information in the text of the manuscript:

To find out in which part of chromatin a gene resides we used the published data of Aldiri et. Al. [31] at https://viz.stjude.cloud/dyer-lab/visualization/the-4-dimensional-nucleome-of-murine-rod-photoreceptors~72. The authors predicted chromatin state of the genome loci of particular genes (euchromatin or heterochromatin) at E14.5 or PN21 developmental stages. As we completed our experiments at PN24, we used the PN21 chromatin predictions from this source. 

  1. Comments 8. Reference 5 should be fixed. The cited reference is a 'News & Views' article that refers to the following paper: PMID: 31964843. I think that this latter citation should be used instead.

Response 8. Thank you very much, we will use proper citation in the revised version of the paper.

9.Comments 9.Kindly include the initial research article that shows DZNep and UNC1999 inhibit EZH2. Include these references in the result section, where you mentioned that the EZH2 inhibitor DZNep...or so. Do the same for the G9a/GLP inhibitor UNC0642.

 Response 9. We have incorporated those citations to the manuscript.

10.Comments 10. The genes in Figures 1B, 2B, 3C, 4A, and 6A are difficult to read out. I understand that achieving readability in some of the figures will be challenging, but please make an effort in the figures where it is feasible.

Response 10. We tried to improve the quality of the figures and included in heatmaps only 100 most changed genes. Additionally, we supplemented heatmaps, with a list of genes that show significant expression changes (FDR<0.05) for each inhibitor/condition (in Supplement table 2).

Reviewer 2 Report

Comments and Suggestions for Authors

In the present studies, the authors have used inhibitors of LSD1 and HDAC1 to show that they decondense chromatin and preserve retinas from degeneration in rd10 mice. They also proposed the metabolic pathways that these inhibitors may use to protect the retina from degeneration. The manuscript is well written, and the research methodology is appropriate enough to draw conclusions. I suggest incorporating some minor corrections into the manuscript to improve its readability for the audience.

  1. Kindly include the initial research article that shows DZNep and UNC1999 inhibit EZH2. Include these references in the result section, where you mentioned that the EZH2 inhibitor DZNep...or so. Do the same for the G9a/GLP inhibitor UNC0642.
  2. The genes in Figures 1B, 2B, 3C, 4A, and 6A are difficult to read out. I understand that achieving readability in some of the figures will be challenging, but please make an effort in the figures where it is feasible.

Author Response

(The authors gave the same response as above.)
